# ABAS-RAL: ADAPTIVE BATCH SIZE USING REINFORCED ACTIVE LEARNING

## ABSTRACT

Active learning reduces annotation costs by selecting the most informative samples, however fixed batch sizes used in traditional methods often lead to inefficient use of resources. We propose Adaptive BAtch Size using Reinforced Active Learning, a novel approach that dynamically adjusts batch sizes based on model uncertainty and performance. By framing the annotation process as a Markov Decision Process, the proposed method employs reinforcement learning to optimize batch size selection, using two distinct policies: one targeting precision and budget, and the other for adapting the batch size based on learning progress. The proposed method is evaluated on both CIFAR-10, CIFAR-100 and MNIST datasets. The performance is measured across multiple metrics, including precision, accuracy, recall, F1-score, and annotation budget. Experimental results demonstrate that the proposed method consistently reduces annotation costs while maintaining or improving performance compared to fixed-batch Active Learning methods, achieving higher sample selection efficiency without compromising model quality.

## 1 INTRODUCTION

Active Learning (AL) reduces the effort required for image classification by selecting the most informative samples for labeling (Zhan et al., 2022). However, labeling large datasets can be a costly and time-consuming process, especially when expert knowledge is required (Mahmood et al., 2022). Additionally, poor-quality annotations can negatively impact machine learning (ML) performance, further increasing the complexity and cost of AL. Reinforcement Learning (RL) (Takezoe et al., 2023; Ren et al., 2020) offers a promising solution to these challenges by dynamically adjusting batch sizes and optimizing sample selection strategies. RL operates by learning from feedback, rewarding actions that enhance performance and efficiency, making it well-suited for sequential decision-making tasks, such as sample selection in AL, where outcomes are uncertain and evolve over time. Despite these advantages, RL has not yet been widely used to optimize batch sizes in AL. Most RL-based methods focus on broader decision-making tasks, leaving a gap in the dynamic adjustment of batch sizes, which could potentially improve AL's efficiency.

Reinforced Active Learning (RAL) (Takezoe et al., 2023; Ren et al., 2020) merges RL with AL by enabling an agent to select the most relevant data based on rewards and penalties. This approach has been successfully applied in fields such as robotics (Kober et al., 2023), natural language processing (NLP) (Fang et al., 2017), and computer vision (Le et al., 2021), where it has improved learning efficiency by focusing on the most useful data. In RAL, an RL agent autonomously replaces traditional AL strategies such as uncertainty-based (Cao & Tsang, 2021; Kim et al., 2020; Sinha et al., 2019), diversity-based (Sener & Savarese, 2017), or hybrid methods (Ash et al., 2019). Rather than relying on fixed rules, the agent dynamically adapts its sample selection based on learned experiences, optimizing the learning process by concentrating on the most informative data.

Although the above-mentioned methods improve AL efficiency, mostly rely on fixed batch sizes or predefined strategies, which can limit their adaptability and resource optimization. The proposed method, Adaptive BAtch Size using Reinforced Active Learning (ABAS-RAL), addresses these limitations by using RL to dynamically adjust batch sizes based on real-time feedback, resulting in more efficient and effective sample selection. The key contributions of this work are:

1. **Adaptive Batch Size Selection:** We introduce a method that dynamically adjusts the batch size based on real-time feedback, optimizing both efficiency and model performance.

2. **Dual-Policy Design:** A single RL agent manages precision and budget with one policy, while another policy determines the optimal batch size for each iteration.

3. **Optimized Learning:** ABAS-RAL enhances learning efficiency by focusing on the most relevant data, reducing annotation costs while maintaining high precision.

## 2 RELATED WORK

### 2.1 ACTIVE LEARNING

AL selects the most informative data points for labeling to enhance learning efficiency and reduce costs, especially when labeled data is scarce. *Uncertainty-based* AL (Cao & Tsang, 2021; Kim et al., 2020; Sinha et al., 2019) targets data points with high uncertainty, while *hybrid* (Ash et al., 2019) and *interpolation-based* (Parvaneh et al., 2022) methods combine strategies or focus on samples near labeled points. *ALFA-Mix* (Parvaneh et al., 2022) and *LOC* (Mahmood et al., 2022) are specific methods that improve AL by evaluating label variability and optimizing data collection to balance costs and performance.

### 2.2 REINFORCED ACTIVE LEARNING

RL has advanced AL by dynamically selecting the most informative samples. Key contributions include Fang et al. (2017) for NLP, Woodward & Finn (2017) for image classification, Pang et al. (2018) for meta-learning, and Sun & Gong (2019) for image selection. Wassermann et al. (2019) introduced contextual-bandit models, Liu et al. (2019) developed a human-in-the-loop model, and Wang et al. (2020) applied RL to medical imaging. Casanova et al. (2020) focused on semantic segmentation, Hanane et al. (2022) on multi-agent RL for medical segmentation, and Ahmad et al. (2021) on selecting informative image regions. Recent work includes Nilsson et al. (2021) on lifelong visual perception, Slade & Branson (2022) on deep RL for medical image classification, Cui et al. (2022) on human activity recognition, Katz & Kravchik (2022) on online stream-based meta AL, Dodds et al. (2023) on molecular design, and Chun (2023) on data-efficient classification frameworks.

### 2.3 ADAPTIVE BATCH SIZE APPROACHES

Research on adaptive batch sizes in AL includes Hacohen et al. (2022) on budget and query strategies, Chakraborty et al. (2014) on optimizing batch size and selection criteria, and Ishibashi & Hino (2017) on error stability-based stopping. Cai et al. (2017) focused on Expected Model Change Maximization to select informative samples. Konyushkova et al. (2018) developed a universal AL strategy using RL and MDP frameworks. Fazakis et al. (2019) combined AL with semi-supervised learning for efficient data utilization. Recent advancements by Haghighatlari et al. (2020) and Fauld (2022) include EMCM (Expected Model Change Maximization) and AB-EMCM for adaptive batch size in regression tasks.

ABAS-RAL utilizes RL to dynamically determine the optimal batch size for visual classification tasks. The proposed method uses RL to adaptively select the batch size in a single, efficient step. This is achieved with minimal additional cost or computational overhead, as RL agent not only selects the batch size but also utilizes the classifier to fill it effectively. This approach ensures that batch sizes are tailored in real-time to maximize learning efficiency and performance, addressing the limitations of existing methods and advancing adaptive batch size techniques in a practical, classification-focused manner.

## 3 ADAPTIVE BATCH SIZE SELECTION USING RL

The introduced RL-based approach uses an agent, called BatchAgent, which learns to select the optimal number of samples to annotate. The BatchAgent operates within a defined state-space, choosing actions (batch sizes) and receiving rewards based on improvements in model performance.

By focusing on informative samples, this method reduces the number of annotations needed for satisfactory performance.

### 3.1 FORMULATING AL AS A MARKOV DECISION PROCESS

Following state-of-the-art approaches (Konyushkova et al., 2018) we use the deep Q-network (DQN) (Mnih et al., 2013) method and modify AL process based on this methodology. Based on the work of Konyushkova et al. (2018), the proposed method also takes advantage of the benefits of Q-learning (Watkins & Dayan, 1992). BatchAgent is trained with the goal of being able to identify an optimal batch of samples at each iteration of an AL episode.

### 3.2 RL STATES

At the beginning of AL training, a subset $V$ (state data) is designated from the unlabeled data pool $U_0$ and then update $U_0$ to exclude $V$, resulting in $U_0 \setminus V$. At each iteration $t$, the state representation is defined as a vector $\mathbf{s}_t$ consisting of the sorted margin scores $m(x_i)$ for each $x_i$ in $V$. The margin scores, computed based on the classifier's predictions for each sample in the state data $V$, are calculated as the difference between the highest and the second-highest predicted class probabilities for each sample of the state data $V$, i.e., $m_i(x_i) = P(y^*|x_i) - P(y^{**}|x_i)$, where $y^*$ and $y^{**}$ are the most likely and the second most likely class labels for sample $x_i$. Sorting the margin scores ensures the input is consistent and structured, allowing RL agent to easily identify the most uncertain samples. This helps the agent to make better learning decisions, as an unsorted input would be harder to interpret and generalize. The state representation captures essential details, like the mean prediction and the level of classifier confidence, and is a simple form derived from the classifier and the dataset.

### 3.3 RL REWARDS CALCULATION

The reward $r_t$ for an action $a_t$ taken at iteration $t$ is defined as:

$$r_t = \left( \frac{P(B_{t+1})}{B_{t+1}} \right) - \left( \frac{P(B_t)}{B_t} \right) \tag{1}$$

where $B_t$ represents the current batch at iteration $t$ for an AL run, and $P(B_t)$ is the precision at iteration $t$. The reward function $r_t$ (Equation 1) measures the improvement in precision per unit of annotation effort. Comparing batch precisions in the reward function helps the agent to choose batch sizes that improve performance. By focusing on calculating the improvement in precision from one batch to the next, the agent learns to pick batch sizes that give the best results with the least effort. This ensures that the model becomes more accurate without wasting too many labeled samples. The latter helps the agent to select efficient annotation strategies by favoring actions that result in significant precision improvements with minimal annotation cost. In other words, equation 1 compares the efficiency of the current total batch $B_{t+1}$ with the previous total batch $B_t$, guiding the agent to choose batch sizes that maximize precision gains per annotated sample.

### 3.4 Q-VALUE UPDATE

Q-values are used in RL to estimate the potential future rewards of taking a specific action in a given state (Watkins & Dayan, 1992). The goal is to maximize these rewards over time. In the context of the suggested model, Q-values help the agent decide the best batch size to choose by predicting which action will lead to the highest long-term improvement in precision. We use a DQN to estimate these Q-values and update them as the model learns.

DQN updates its Q-values by minimizing the difference between the predicted and the target Q-values. The target Q-value for a state-action pair is given by:

$$Q_{\text{value}} = r_t + \gamma \times \max_a Q_{\text{target}}(s_{t+1}, a) \tag{2}$$

where $r_t$ represents the reward obtained after taking action $a_t$ in state $s_t$. The term $\gamma$ is the discount factor, which reflects the importance of future rewards in the current decision-making process. The

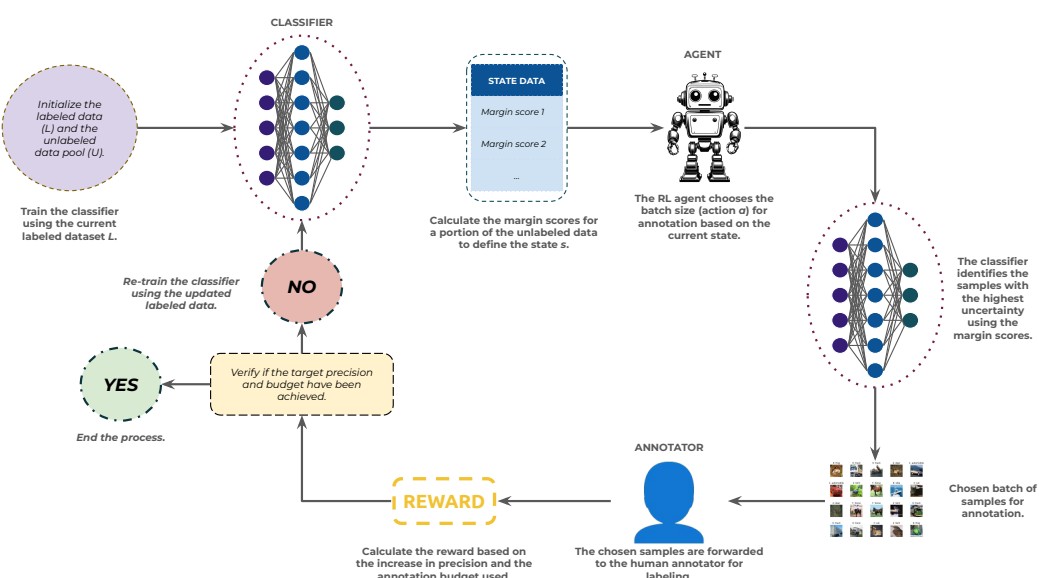

Figure 1: Flowchart of RL agent's decision process in ABAS-RAL.

expression $Q_{\text{target}}(s_{t+1}, a)$ refers to the maximum Q-value associated with the next state $s_{t+1}$, as determined by the target network, over all possible actions $a$ that could be taken in that state.

### 3.5 RL ACTION SELECTION

DQN selects actions that maximize the Q-value, guiding the agent to choose batch sizes that yield the highest cumulative reward. By updating Q-values based on received rewards, DQN optimizes the annotation process for efficient and effective learning.

At each iteration $t$, the action $a_t$ is defined as selecting $k$ samples for annotation, where $k \in \{1, 2, \ldots, \lfloor 0.1 \times |U_t| \rfloor\}$, and $|U_t|$ is the total number of available unlabeled samples. Limiting the action space to 10% of the dataset per iteration encourages thorough data exploration, prevents overfitting, and ensures efficient use of annotation resources. This approach also provides more frequent feedback, helping the agent to quickly identify informative samples and refine the model iteratively.

### 3.6 RL AGENT DECISION PROCESS

To better illustrate the decision-making process of RL agent, Fig. 1 provides a flowchart that outlines each step from state representation to reward calculation. RL agent begins by training the classifier on the currently labeled dataset $L$. It then evaluates the uncertainty of samples in the unlabeled pool $U$ to represent the current state $s$. Based on this state, the agent selects an optimal batch size (*action a*) for annotation. The most uncertain samples are then forwarded to the human annotator for labeling, after which the datasets are updated. The agent receives a reward based on the improvement in precision relative to the annotation budget, guiding future batch size selections.

### 3.7 WARM-START EPISODES

Warm-start episodes are crucial for initializing the AL algorithm and training the BatchAgent. During these episodes, the BatchAgent tests different batch sizes to learn their effects on performance. An episode ends when:

- Two consecutive rewards decrease, or
- All samples have been used.

Two consecutive reward decreases signal the end of an episode because it shows the model is no longer improving. Stopping here avoids wasting time on ineffective batches, allowing the model to explore new ones and use them to better train DQN agent. The details of the warm-start episode process are outlined in Algorithm 1.

Following these warm-start episodes, key benchmarks are set:

- *Target Precision*: The highest precision achieved during warm-start episodes.
- *Target Budget*: The minimum number of samples required to achieve this precision.

These benchmarks help guide the BatchAgent's training, ensuring it can maintain or exceed the target precision efficiently while staying within resource limits.

---

**Algorithm 1** Warm-Start Episode Process

---

**Require:** $L$ labeled data, $D$ training set, $b$ batch, $c$ classifier, $r$ reward, $s$ state
**Ensure:** Trained classifier
  1: Randomly choose the first $b$ of samples from $D$ and train $c$.
  2: **repeat**
  3:     Fill the selected $b$ with random samples.
  4:     Compute the $s$ (margin scores for the state data) using $c$.
  5:     Re-train the $c$ with the new $L$.
  6:     Compute the $r$ based on the precision of the $c$ and the selected $b$ size.
  7:     Store current $s$, selected $b$, and $r$ in the Replay Buffer.
  8: **until** termination condition is met when two consecutive rewards decrease or the samples are exhausted.
  9: **return** Trained $c$, Target precision, Target budget

---

### 3.8 POLICY

The BatchAgent's policy determines the batch size for each annotation episode, aiming to balance exploration of new batch sizes with exploitation of existing knowledge to maximize precision. An episode concludes when:

- **Target Precision** is achieved, ensuring optimal performance without over-training or under-training.
- **Target Budget** is reached, keeping annotation within resource constraints.

This approach ensures that the BatchAgent operates effectively within budget limits while striving for high precision.

### 3.9 PROCEDURE

As described in Algorithm 2, the BatchAgent is trained with the goal of being able to identify an optimal batch of samples at each iteration of an AL episode. Specifically, during model training, a predefined number of epochs are executed, consisting of a predefined number of episodes.

These steps repeat until a *terminal state* $s_T$ is reached. Using the *best precision score $p$* as the best precision score that was achieved from the warm-start episodes, the *target precision* is defined. Using the smallest achieved budget from the warm-start episodes, the *target budget* is defined. Thus, the *terminal state* $s_T$ is reached when the *target precision* and the *target budget* are achieved.

By selecting and labeling samples, updating the classifier, and calculating rewards in each step, the BatchAgent improves its performance and adjusts to changes in the model's predictions.

### 3.10 DQN IMPLEMENTATION DETAILS

RL is implemented using non-linear Q-function approximation inspired by DQN (Watkins & Dayan, 1992) and incorporate key techniques for effective learning:

---

**Algorithm 2** BatchAgent Training Algorithm

---

**Require:** $L_0$ initial labeled data, $U_0$ initial unlabeled data, $D$ dataset, $\pi$ policy, Target precision $P_{target}$, Target budget $B_{target}$, $E$ training epochs, $T$ episodes per epoch
**Ensure:** Trained agent $DQN_T$, Trained classifier $c_T$
 1: **for** $e = 0$ to $E - 1$ **do** ▷ For each epoch.
 2:   **for** $t = 0$ to $T - 1$ **do** ▷ For each episode.
 3:     **repeat**
 4:       Train classifier $c_t$ using labeled data $L_t$.
 5:       Characterize state $s_t$ based on $c_t$.
 6:       Select action $a_t \in A$ following policy $\pi : s_t \to a_t$ to define batch $X_t \subseteq U_t$.
 7:       Retrieve labels $Y_t$ for $X_t$ from $D$.
 8:       Update labeled set $L_{t+1} = L_t \cup (X_t, Y_t)$.
 9:       Update unlabeled set $U_{t+1} = U_t \setminus X_t$.
10:       Use classifier $c_t$ to fill batch action based on uncertainty scores.
11:       Assign reward $r_{t+1}$ based on empirical performance.
12:       Update agent's $DQN_t$ weights.
13:     **until** Precision $P_t \geq P_{target}$ **and** Budget $B_t \leq B_{target}$ **or** the samples are exhausted.
14:   **end for**
15: **end for**
16: **return** $DQN_T, c_T$

---

- **Target Network:** A separate target network with a slow update rate (0.01) (Watkins & Dayan, 1992) stabilizes learning by decoupling it from the main Q-network.

- **Replay Buffer:** A buffer of size 50,000 (Watkins & Dayan, 1992) mitigates correlated updates by providing diverse data for training.

- **Double DQN:** Used to reduce overestimation bias in Q-value estimates by separating action selection from value estimation (Watkins & Dayan, 1992).

- **Prioritized Experience Replay:** Experiences are prioritized based on temporal-difference errors (Watkins & Dayan, 1992), with the prioritization exponent set to 3 to balance exploration and prioritization.

Additionally, the bias of the final neural network layer is initialized to the average reward from warm-start episodes, which helps to stabilize learning in environments with negative rewards. State representation is preprocessed for compatibility with fully connected layers, and scores within the state are sorted. The network uses sigmoid activations in the final layer, while the output layer employs a linear activation to estimate Q-values.

## 4 EXPERIMENTAL EVALUATION

To evaluate the performance of **ABAS-RAL**, a series of experiments is conducted comparing it with **Random Sampling (RS)** and several **fixed-batch size methods** on three datasets: CIFAR-10, CIFAR-100 (Alex & Hinton, 2009), and MNIST (Lecun et al., 1998). Each method is tested using 20% of the training set as evaluation data. For every method and dataset, 50 experiments are performed, recording precision, accuracy, recall, F1-score, and the annotation budget (expressed as a percentage of the evaluation samples).

The experiments are organized into three main comparisons across all datasets:

- **ABAS-RAL vs. Random sampling:** Both methods are tested using the same pre-trained ResNet50 (He et al., 2016) classifiers for each dataset.

- **ABAS-RAL vs. Fixed-batch size methods:** ABAS-RAL is compared with Entropy Sampling (ES), Uncertainty Sampling (US), and ALFA-Mix (Parvaneh et al., 2022). The same pre-trained ResNet50 classifiers are used for all comparisons for each dataset.

- **Training from scratch:** ABAS-RAL and the fixed-batch methods are further evaluated by training the classifiers from scratch. For all datasets (CIFAR-10, CIFAR-100, and MNIST), ResNet50 is used. Each method aims to achieve 85% of the target precision.

## 4.1 PARAMETERS

- **Dataset:** Each dataset (CIFAR-10, CIFAR-100, MNIST) is split randomly, maintaining an equal number of samples per class. The 10% of each dataset's training data is used as state data, 10% for warm-start episodes, 60% for DQN training, and 20% for evaluation. This split is consistent across all methods to ensure fair comparisons.

- **BatchAgent:** 50 epochs are executed with 5 episodes per epoch and 100 neural network updates.

- **Classifier:**

  - **CIFAR-10:** Pre-trained ResNet50 on ImageNet, modified for 32x32x3 input, global average pooling, and a 10-class softmax layer. The classifier is trained for 10 epochs, with a learning rate of 0.01 and a batch size of 64.

  - **CIFAR-100:** Pre-trained ResNet50 on ImageNet, modified for 32x32x3 input, global average pooling, and a 100-class softmax layer. The classifier is trained for 10 epochs, with a learning rate of 0.01 and a batch size of 64.

  - **MNIST:** Pre-trained ResNet50 on ImageNet, modified for 32x32x3 input, global average pooling, and a 10-class softmax layer. MNIST images are converted to RGB and resized to 32x32. The classifier is trained for 10 epochs, with a learning rate of 0.01 and a batch size of 64.

- **Metrics:** Performance is evaluated using weighted average precision, accuracy, recall, F1-score, and annotation budget (as a percentage of the evaluation samples).

## 4.2 ABAS-RAL VS. RANDOM SAMPLING

ABAS-RAL is compared against RS across the CIFAR-10, CIFAR-100, and MNIST datasets. ABAS-RAL consistently outperforms RS in terms of precision, while using significantly fewer labeled samples.

The results of the 50 experiments, summarized in Table 1, show that, for CIFAR-10, CIFAR-100, and MNIST, ABAS-RAL achieves comparable or higher perfomance metrics, but with a far smaller annotation budget compared to RS. These results highlight ABAS-RAL's advantage over RS in both effectiveness and cost-efficiency.

Table 1: Comparison of ABAS-RAL and RS results across 50 experiments on CIFAR-10, CIFAR-100, and MNIST. Each experiment involves using the pre-trained ResNet50 classifier (from DQN agent's training phase). The budget represents the percentage of samples used from the total evaluation data.

| Dataset: CIFAR10 — *Target Precision: 44.66%, Target Budget: 98.03%* | | | | | | |
|---|---|---|---|---|---|---|
| | | Precision | Accuracy | Recall | F1-Score | Budget |
| ABAS-RAL | Mean | **46.17%** | 43.47% | 43.47% | 42.67% | **5.12%** |
| | Max | **50.00%** | 45.26% | 45.26% | **45.24%** | 11.17% |
| RS | Mean | 45.89% | **44.00%** | **44.00%** | **42.97%** | 50.79% |
| | Max | 47.85% | **45.62%** | 45.62% | 45.08% | 97.49% |
| Dataset: CIFAR100 — *Target Precision: 28.01%, Target Budget: 99.94%* | | | | | | |
| | | Precision | Accuracy | Recall | F1-Score | Budget |
| ABAS-RAL | Mean | **32.02%** | 32.14% | 32.14% | 31.54% | **1.98%** |
| | Max | **32.52%** | 32.53% | 32.55% | 31.93% | **1.98%** |
| RS | Mean | 31.89% | **32.47%** | **32.47%** | **31.68%** | 48.71% |
| | Max | 32.47% | **33.02%** | **33.02%** | **32.21%** | 99.22% |
| Dataset: MNIST — *Target Precision: 94.11%, Target Budget: 96.34%* | | | | | | |
| | | Precision | Accuracy | Recall | F1-Score | Budget |
| ABAS-RAL | Mean | **95.11%** | **95.05%** | **94.98%** | **95.04%** | **7.07%** |
| | Max | **95.23%** | **95.24%** | **95.24%** | **95.23%** | 8.18% |
| RS | Mean | 95.02% | 95.01% | 94.90% | 95.02% | 60.13% |
| | Max | 95.21% | 95.20% | 95.20% | 95.19% | 95.15% |

Figure 2 shows the batch sizes selected by DQN model, revealing the model's preference for certain sizes that it has learned to associate with higher expected returns for CIFAR-10, CIFAR-100 and MNIST.

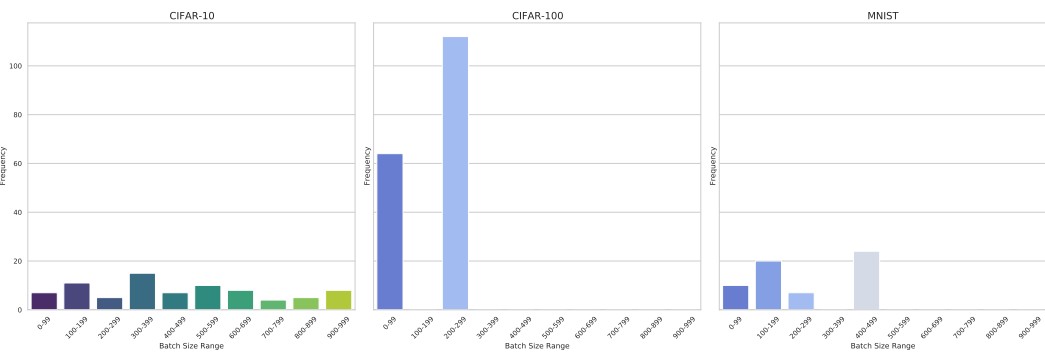

Figure 2: Distribution of batch sizes selected by ABAS-RAL agent across 50 experiments, grouped into 100-intervals for CIFAR-10, CIFAR-100 and MNIST. The histogram shows the frequency of selections within each range, highlighting the agent's preference for certain batch sizes during AL.

### 4.3 ABAS-RAL vs. Fixed-batch size methods

The comparisons between ABAS-RAL and fixed-batch size methods, including ES, US, and ALFA-Mix, are conducted across the CIFAR-10, CIFAR-100, and MNIST datasets, using the same pre-trained classifier from the DQN agent's training phase.

As shown in Table 2, ABAS-RAL consistently outperforms the fixed-batch size methods across all datasets, achieving higher performance metrics while using fewer labeled samples. Its dynamic batch size adjustment allows for more efficient use of the annotation budget, particularly in CIFAR-10 and CIFAR-100, where ABAS-RAL significantly reduces the required budget. Overall, ABAS-RAL proves to be both highly effective and resource-efficient compared to the fixed-batch approaches.

Table 2: Comparison of ABAS-RAL with fixed-batch size methods using the same (pretrained) classifier from the BatchAgent's training phase.

| CIFAR10 — *Target Precision: 44.66%, Target Budget: 98.03%* | | Precision | Accuracy | Recall | F1-Score | Budget |
|---|---|---|---|---|---|---|
| **ABAS-RAL** | **Mean** | **46.17%** | **43.47%** | **43.47%** | **42.67%** | **5.12%** |
| | **Max** | **50.00%** | **45.26%** | **45.26%** | **45.24%** | 11.17% |
| **ES** | Mean | 45.04% | 42.96% | 42.95% | 42.17% | 11% |
| | Max | 47.36% | 44.14% | 44.13% | 43.67% | **11%** |
| **US** | Mean | 46.04% | 43.96% | 43.96% | 43.17% | 11% |
| | Max | 48.36% | 45.14% | 45.14% | 44.67% | **11%** |
| **ALFA-Mix** | Mean | 45.67% | 43.54% | 43.92% | 42.08% | 11% |
| | Max | 47.44% | 44.56% | 45.13% | 44.32% | **11%** |
| CIFAR100 — *Target Precision: 28.01%, Target Budget: 99.94%* | | Precision | Accuracy | Recall | F1-Score | Budget |
| **ABAS-RAL** | **Mean** | **32.02%** | **32.14%** | **32.14%** | **31.54%** | **1.98%** |
| | **Max** | **32.52%** | **32.53%** | **32.55%** | **31.93%** | **1.98%** |
| **ES** | Mean | 31.64% | 32.02% | 32.01% | 31.23% | 11% |
| | Max | 32.41% | 32.11% | 32.13% | 31.43% | 11% |
| **US** | Mean | 31.73% | 32.05% | 32.05% | 31.31% | 11% |
| | Max | 32.05% | 32.30% | 32.31% | 31.53% | 11% |
| **ALFA-Mix** | Mean | 31.82% | 32.29% | 32.29% | 31.53% | 11% |
| | Max | 32.43% | 32.47% | 32.47% | 31.57% | 11% |
| MNIST — *Target Precision: 94.11%, Target Budget: 96.34%* | | Precision | Accuracy | Recall | F1-Score | Budget |
| **ABAS-RAL** | **Mean** | **95.11%** | **95.05%** | **94.98%** | **95.04%** | **7.07%** |
| | **Max** | **95.23%** | **95.24%** | **95.24%** | **95.23%** | 8.18% |
| **ES** | Mean | 95.10% | 94.09% | 94.95% | 94.99% | 7.76% |
| | Max | 95.22% | 95.17% | 95.23% | 95.20% | **7.76%** |
| **US** | Mean | 95.12% | 94.13% | 94.89% | 94.98% | 7.76% |
| | Max | 95.22% | 95.15% | 95.20% | 95.20% | **7.76%** |
| **ALFA-Mix** | Mean | 95.01% | 94.97% | 94.79% | 94.95% | 7.76% |
| | Max | 95.10% | 95.10% | 95.08% | 95.08% | **7.76%** |

## 4.4 EXPERIMENTS WITH CLASSIFIER TRAINING FROM SCRATCH

Table 3 presents the results of training classifiers from scratch, comparing ABAS-RAL with fixed-batch size methods (ES, US, and ALFA-Mix) across CIFAR-10, CIFAR-100, and MNIST datasets. Each experiment aims to reach 85% of the target precision for each classifier and dataset.

In all cases, ABAS-RAL demonstrates better resource efficiency by achieving comparable performance metrics with a lower overall cost. These results confirm that ABAS-RAL is more efficient in terms of both performance and resource use across the datasets.

Table 3: Performance comparison of ABAS-RAL and fixed-batch size methods (ES, US, ALFA-MIX) when training the classifier from scratch.

| | | Precision | Accuracy | Recall | F1-Score | Budget |
|---|---|---|---|---|---|---|
| **CIFAR10 — *Target Precision: 37.96%*** | | | | | | |
| **ABAS-RAL** | Mean | **39.69%** | **36.17%** | **36.17%** | **34.52%** | **7.74%** |
| | Max | **44.73%** | **39.53%** | **39.53%** | **38.46%** | **47.67%** |
| **ES** | Mean | 35.30% | 28.35% | 28.24% | 26.08% | 100% |
| | Max | 43.35% | 32.96% | 32.96% | 32.44% | 100% |
| **US** | Mean | 36.30% | 29.35% | 29.24% | 27.08% | 100% |
| | Max | 44.35% | 33.96% | 33.96% | 33.43% | 100% |
| **ALFA-MIX** | Mean | 34.30% | 27.35% | 29.24% | 27.24% | 100% |
| | Max | 42.35% | 31.96% | 31.44% | 31.96% | 100% |
| **CIFAR100 — *Target Precision: 23.81%*** | | | | | | |
| **ABAS-RAL** | Mean | **25.60%** | **25.73%** | **25.73%** | **24.57%** | **1.98%** |
| | Max | **27.20%** | **27.17%** | **27.17%** | **26.11%** | **1.98%** |
| **ES** | Mean | 24.11% | 25.14% | 24.65% | 23.08% | 70% |
| | Max | 25.43% | 25.96% | 26.12% | 23.44% | 92% |
| **US** | Mean | 24.12% | 25.14% | 24.71% | 23.23% | 71% |
| | Max | 25.50% | 26.65% | 24.91% | 23.43% | 91% |
| **ALFA-MIX** | Mean | 22.55% | 23.84% | 23.34% | 21.98% | 61% |
| | Max | 23.35% | 23.96% | 23.84% | 22.12% | 72% |
| **MNIST — *Target Precision: 79.99%*** | | | | | | |
| **ABAS-RAL** | Mean | **81.85%** | **81.65%** | **80.97%** | **80.91%** | **26.67%** |
| | Max | **82.97%** | **82.91%** | **81.25%** | **81.24%** | **29.18%** |
| **ES** | Mean | 80.87% | 79.23% | 79.02% | 78.01% | 34.10 % |
| | Max | 81.23% | 79.46% | 79.52% | 79.06% | 37.10 % |
| **US** | Mean | 80.20% | 79.30% | 79.50% | 78.90% | 32.10% |
| | Max | 81.00% | 80.00% | 80.30% | 79.70% | 39.10% |
| **ALFA-MIX** | Mean | 80.21% | 80.71% | 80.71% | 80.47% | 40.32% |
| | Max | 81.42% | 81.40% | 81.11% | 81.21% | 41.40% |

As shown in Fig. 3, for each dataset, DQN agent consistently meets precision targets while minimizing annotation costs. The histograms with KDE (Kernel Density Estimate) curves show how DQN agent optimizes learning through budget-efficient decisions.

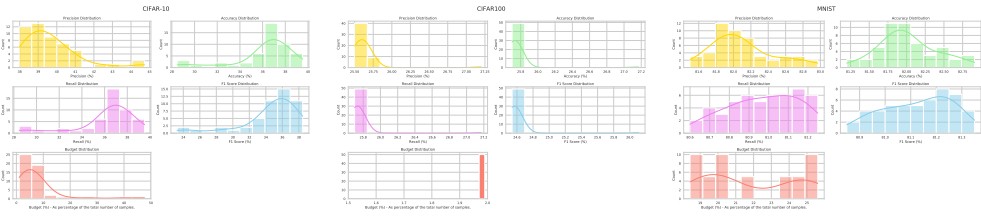

Figure 3: Distribution of metrics for ABAS-RAL for training the classifier from scratch for CIFAR-10, CIFAR-100 and MNIST.

## 4.5 DISCUSSION

The experimental results demonstrate that ABAS-RAL is an efficient AL method across various datasets. Key findings are:

1. **Precision and Cost Efficiency**: ABAS-RAL consistently achieves precision scores comparable to or higher than other methods like RS and fixed-batch approaches. Importantly,

it requires significantly less annotation budget across CIFAR-10, CIFAR-100, and MNIST, making it particularly useful when data annotation is expensive. ABAS-RAL's ability to maintain high performance with fewer labeled samples highlights its efficiency.

2. **Dynamic Batch Size**: Unlike fixed-batch methods (RS, US, and ALFA-Mix), which often require a large portion of the dataset, ABAS-RAL dynamically adjusts batch sizes based on the model's learning needs. This flexibility allows ABAS-RAL to reach target performance without excessive annotation, making it more resource-efficient.

In all experiments, including CIFAR-10, CIFAR-100, and MNIST, ABAS-RAL outperforms other approaches, particularly in scenarios with limited annotation budgets. Its ability to maintain high precision while minimizing costs makes ABAS-RAL an ideal choice for AL tasks that demand both performance and resource efficiency.

### 4.6 SYSTEM SETUP

The ResNet50 model, pre-trained on ImageNet (TensorFlow 1.14, Python 3.6), is used to evaluate the CIFAR-10, CIFAR-100, and MNIST datasets. The classifier achieves precisions of approximately 45% for CIFAR-10, 29% for CIFAR-100, and 95% for MNIST when trained for 10 epochs on the full training set of each dataset. All experiments were conducted on an NVIDIA GTX 1080 Ti GPU.

## 5 CONCLUSION AND FUTURE WORK

In this study, we introduce ABAS-RAL, a method for optimizing data sampling in AL that improves precision while efficiently managing annotation budgets. By utilizing RL, ABAS-RAL enhances model performance and resource utilization, proving effective in resource-constrained settings. Future work could explore its application to diverse datasets and real-world scenarios to test its robustness and practical effectiveness. Additionally, investigating hybrid approaches that combine RL with other AL strategies and incorporating domain-specific knowledge could further refine ABAS-RAL's adaptability and performance across various tasks.

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
