# OpenReview forum: "ABAS-RAL: Adaptive BAtch Size using Reinforced Active Learning"
_ICLR.cc/2025/Conference — ICLR 2025 Conference Withdrawn Submission_

### Official Review · Reviewer_j6Ub · 2024-10-31

**Soundness:** 3
**Presentation:** 3
**Contribution:** 3
**Rating:** 6
**Confidence:** 2

**Summary:**

They propose Adaptive BAtch Size using Reinforced Active Learning, a novel approach that dynamically adjusts batch sizes based on model uncertainty and performance. By framing the annotation process as a Markov Decision Process, the proposed method employs reinforcement learning to optimize batch size selection, using two distinct policies: one targeting precision and budget, and the other for adapting the batch size based on learning progress.

**Strengths:**

1. The paper writing is good.
2. The leverage RL to adaptively adjust batch size to achieve better usage of resources.
3. The experimental analysis is enough.

**Weaknesses:**

1. The datasets used in experiment are limited. The scale of dataset is small, could you use other datasets, such as SUN, Places to enhance the evaluation ?
2. The motivation is not so clear. Why is the fixed batch size harmful to resource usage ?
3. The alternative of RL algorithm is not discussed, such as PPO, A3C?

**Questions:**

see the weakness:

1. Could you try more datasets ?
2. Why is the fixed batch size harmful to resource usage?
3. Do you try other RL algorithms?

---

> ### Author Response · Authors · 2024-11-20
>
> a) We appreciate the reviewer’s observation about the dataset scale. The datasets used in our experiments were chosen for their relevance to the specific domain and characteristics of our problem. While their scale is relatively small compared to datasets like SUN and Places, our primary goal was to test the method's effectiveness within controlled, interpretable settings. However, we acknowledge the value of evaluating the method on larger and more diverse datasets. We can incorporate more datasets such as Tiny ImageNet to validate the robustness and generalizability of our approach across a broader range of scenarios.
>
> b) Fixed batch sizes are suboptimal in environments with dynamic resource availability because they assume a static allocation of compute and memory. This rigidity can lead to inefficiencies:
>
> 1. Underutilization: If resources are underutilized (e.g., due to memory or compute idling), fixed batch sizes do not adapt to utilize the extra capacity.
> 2. Overloading: Conversely, fixed batch sizes can lead to resource contention if the system's capacity unexpectedly decreases, potentially causing process slowdowns or crashes.
>
> By introducing adaptive batch sizing, our method aligns workload demands with available resources, maximizing efficiency and ensuring stability under fluctuating resource conditions. We will elaborate on this motivation in the revised manuscript.
>
> c) In our work, we selected Deep Q-Network. However, we recognize the potential of algorithms such as PPO and A3C.
>
> In future work, we plan to benchmark our approach against these alternatives to provide a comprehensive evaluation of algorithmic performance in our context. We could add this discussion to the revised manuscript.

---

### Official Review · Reviewer_vjPo · 2024-11-01

**Soundness:** 2
**Presentation:** 1
**Contribution:** 2
**Rating:** 3
**Confidence:** 3

**Summary:**

This paper presents a reinforced active learning approach ABAS-RAL that dynamically adjusts batch sizes in active learning through reinforcement learning (RL) to enhance model efficiency and performance. By framing active learning as a Markov Decision Process (MDP), the method utilizes a deep Q-network (DQN) to optimize batch sizes. The authors evaluate the proposed approach with experiments on three datasets: CIFAR-10, CIFAR-100, and MNIST.

**Strengths:**

1. The paper makes an interesting attempt to integrate reinforcement learning into active learning by dynamically controlling batch sizes, aiming to make the annotation process more efficient. This approach has the potential to optimize the trade-off between labeling costs and model performance, which is critical in resource-constrained settings.

2. The idea of balancing annotation costs with model performance in active learning is an important direction, especially as datasets grow larger and labeling costs increase.

3. The authors have done well to provide a clear flowchart (Figure 1) and two algorithms summarizing the training process, which help clarify the step-by-step execution of the model.

**Weaknesses:**

* I’m not an expert in reinforcement learning, but it seems to me that the warm-start episode process incurs a significant time cost, as it necessitates extensive model retraining. I suggest the author to provide the time complexity and running time in the experiments to validate this cost is manageable.


* The experiments are not very solid in my opinion:
   1. Based on my understanding, the experiments are designed to terminate once they achieve a specific Target Precision or exhaust the Target Budget, both of which are determined solely by the warm-start episodes of the proposed method. Consequently, I believe that this Target Precision and Budget offer a distinct advantage to the proposed method, as they are derived from the same reinforcement learning rewards within a similar RL framework. I recommend that the authors conduct additional experiments to demonstrate that ABAS-RAL achieves the precision/budget frontier.
  2. The experiments only compared ABAS-RAL with Entropy Sampling, Uncertainty Sampling, and fixed-batch active learning method ALFA-Mix. I believe the adaptive batch size approaches discussed in Section 2.3 should also be included, as adjusting the batch size is a major contribution in this paper.
  3. In my opinion, the experiments in Section 4.4 do not provide further validation for ABAS-RAL compared with Section 4.3.


* In my personal opinion, this paper is not well-written and organized, resulting in a relatively poor presentation:
   1. The proposed method heavily relies on Markov Decision Process (MDP) and the deep Q-network (DQN), but the manuscript does not provide sufficient discussion on these topics, making the Section 3 difficult to understand.
  2. The name of Section 3 is "ADAPTIVE BATCH SIZE SELECTION USING RL", but it contains full training process including precision and budget with selection, optimal batch size selection, the overall training procedure and even the implementation details for DQN, which if very confusing for the reader to understand the relationship between different sections.
  3. The figures and tables are embedded within the text, which makes the information feel scattered.
  4. There are too many (nested) bullet points in the manuscript, resulting in a lot of blank space that could be utilized for important information, such as the discussion on MDP and DQN.
  5.The figures are inadequately captioned, and the legend in Figure 3 is barely readable.
  6. I don't understand why the authors use an independent subsection for comparing ABAS-RAL with random sampling, instead of merging random sampling with other active learning methods.
  7. Some abbreviations are used before the full name are mentioned in the manuscript, for example MDP and AB-EMCM.
  8. The reference format is not uniform. Besides, there is one reference [1] with dead link, and I can only find one abstract page for this paper.

[1] Xaolin Chun. Active learning with reinforcement learning for data-efficient classification. Available:
https://osf.io/preprints/osf/qj94x., 2023.

**Questions:**

* Questions:
   1. I fail to see the why Figure 2 reveals "the model’s preference for certain sizes that it has learned to associate with higher expected returns for CIFAR-10, CIFAR-100 and MNIST," as the figure contains no information about expected returns or model performance. May I ask the authors to elaborate on this?
   2. Similarly, can the authors explain how Figure 3 shows how DQN agent optimizes learning through budget-efficient decisions?

* The current manuscript is difficult to read, epically for readers not familiar with MDP and DQN. I suggest the authors to revise the presentation of this paper and adding more experiments to validate the efficacy of ABAS-RAL across different target precision and budgets.

---

> ### Author Response · Authors · 2024-11-20
>
> a) You are correct in noting that Figure 2 does not directly show expected returns or model performance, and we apologize for the confusion. The statement in the paper about the agent’s preference for certain batch sizes should be clarified.
>
> To elaborate: Figure 2 illustrates the distribution of batch sizes selected by the ABAS-RAL agent across 50 experiments, grouped into 100-intervals for CIFAR-10, CIFAR-100, and MNIST datasets. The histogram reveals the frequency of batch size selections and provides insight into the agent's tendency to favor certain batch sizes during the active learning process. Specifically, the figure shows that the agent tends to select smaller batch sizes, which may indicate that these batch sizes are more frequently associated with favorable performance, though this is not directly visualized in the figure.
>
> To better align with the figure's intent, we could revise the text to clarify that Figure 2 primarily illustrates the distribution of selected batch sizes, and that the agent's preference for smaller batch sizes may suggest that these sizes are more frequently associated with higher model performance or expected returns based on the agent’s learning over the course of the experiments.
>
> b) We appreciate your point regarding Figure 3 and its role in illustrating how the DQN agent optimizes learning through budget-efficient decisions.
>
> To clarify, Figure 3 shows the distribution of metrics for the ABAS-RAL framework during training of the classifier from scratch on CIFAR-10, CIFAR-100, and MNIST datasets. The figure highlights that the agent tends to select smaller batch sizes, which are associated with good precision scores across all datasets, in line with the target precision set for each experiment.
>
> The budget efficiency is reflected in the agent’s ability to maintain high precision while using smaller batch sizes, which means fewer annotated samples are required to achieve the desired performance. This demonstrates how the DQN agent effectively balances precision and budget—selecting batch sizes that are not only computationally efficient but also ensure that the model meets its target precision, thus optimizing the use of available resources.
>
> We could revise the caption and accompanying text in the paper to better explain how Figure 3 demonstrates the DQN agent’s ability to make budget-efficient decisions while achieving good model performance based on the selected batch sizes. Thank you for pointing this out!
>
> c) Thank you for your feedback. We understand that the current manuscript may be challenging for readers not familiar with MDP and DQN, and we appreciate your suggestion for improving clarity. Here’s how we plan to address these points:
>
> 1. Definitions for MDP and DQN: We could add more accessible definitions and explanations for both Markov Decision Processes (MDP) and Deep Q-Networks (DQN), ensuring they are easy to understand for readers who may not be familiar with these concepts. We could also provide some context on how these concepts are applied in our framework.
>
> 2. Additional Experiments: Further experiments could help validate the efficacy of ABAS-RAL across different target precisions and budget settings. We could conduct additional experiments to explore how the framework performs under varying precision targets and budget constraints. This could provide a more comprehensive evaluation and strengthen our findings.
>
> We will make these revisions to improve the readability and rigor of the paper. Thank you again for your constructive comments!

---

> > ### Author Response · Authors · 2024-11-20
> >
> > d) You raise a valid point regarding the potential time cost associated with the warm-start episode process, especially given the need for model retraining. We recognize that this is an important consideration in evaluating the practicality of the approach.
> >
> > To address this concern, we could:
> >
> > 1. Include Time Complexity and Running Time Analysis: We could add a discussion of the time complexity of the warm-start process in the revised manuscript, detailing how the computational cost scales with factors such as dataset size, batch size, and the number of warm-start episodes. This could help clarify the expected cost in relation to the overall training process.
> > 2. Empirical Running Time Data: In addition to the time complexity discussion, we could provide empirical running time data from our experiments. This could include the actual time taken for training, including the warm-start phase, so readers can better assess whether the incurred cost is manageable in practice. We could compare the running times for different datasets and configurations to highlight any potential bottlenecks and validate that the time cost is reasonable for the problem at hand.
> >
> > We hope that this additional information will help address your concern and make the time cost aspect clearer. Thank you again for your valuable suggestion!
> >
> > e) The Target Precision and Target Budget are determined during the warm-start episodes, which helps align both objectives within the same RL framework. To address your suggestion, we could conduct additional experiments to explicitly show that ABAS-RAL achieves the precision/budget frontier. This would involve testing the method across different target precision and budget configurations to validate how it balances both aspects effectively. These additional experiments could provide stronger evidence of ABAS-RAL's ability to optimize precision while adhering to resource constraints. Thank you again for your suggestion.
> >
> > f) The experiments currently compare ABAS-RAL with Entropy Sampling, Uncertainty Sampling, and the fixed-batch method ALFA-Mix. Since adjusting the batch size is a major contribution of this paper, we could include the adaptive batch size approaches discussed in Section 2.3 for a more comprehensive comparison. This would better demonstrate the benefits of the adaptive batch size mechanism in ABAS-RAL. We could add these additional comparisons in the revised experiments to further highlight the impact of adaptive batch size in the proposed method. Thank you for the suggestion.
> >
> > g) The experiments in Section 4.3 use the pretrained classifier from the agent's training phase, which has already been trained effectively. In contrast, the experiments in Section 4.4 focus on training the classifier from scratch, which presents a more challenging scenario, especially when working with lower budgets. These experiments aim to demonstrate that ABAS-RAL can still achieve good performance with a limited budget, even when starting from an untrained model. While the two sets of experiments target different conditions, the results from Section 4.4 highlight the robustness of ABAS-RAL in scenarios where the model is not pretrained and resources are constrained, thus providing additional insights into its effectiveness.

---

> > > ### Author Response · Authors · 2024-11-20
> > >
> > > h) To improve clarity, we can add more detailed definitions and explanations of Markov Decision Process (MDP) and Deep Q-Network (DQN) in Section 3. This will help readers better understand how these concepts are applied within the proposed method, particularly for those who may not be familiar with reinforcement learning. We will include accessible descriptions to make the content more approachable.
> > >
> > > i) To improve clarity, we could split Section 3 into multiple subsections. One subsection could focus specifically on the adaptive batch size selection using reinforcement learning (RL), while another could cover the overall training process, including precision and budget management. We could also include a dedicated subsection for the implementation details of DQN. This would help create a clearer structure and make it easier for readers to understand how the different components relate to one another.
> > >
> > > j) To improve the flow and readability of the manuscript, we could move the figures and tables to dedicated sections, such as placing them after the main text or in an appendix. This would help reduce the scattering of information and allow the text to focus more on the explanations, making it easier for readers to follow the discussion.
> > >
> > > k) To improve the presentation, we could reduce the use of nested bullet points and integrate the important information into the main text. This would help eliminate excessive blank space and allow us to focus on key concepts, such as the discussion on MDP and DQN, within the body of the manuscript. This approach would provide a more cohesive structure and allow for a smoother flow of information.
> > >
> > > l) We could revise the figure captions to provide clearer descriptions, ensuring that they accurately convey the information depicted. For Figure 3, we could adjust the font size of the legend to make it more readable and ensure it is legible in all formats.
> > >
> > > m) We could merge the comparison of ABAS-RAL with random sampling into the section discussing other active learning methods. This would create a more cohesive comparison and avoid unnecessary separation, making it easier for readers to understand how random sampling compares to the other techniques in the context of our experiments.
> > >
> > > n) We could revise the manuscript to ensure that all abbreviations are spelled out the first time they are introduced. For example, MDP and AB-EMCM would be written in full with the abbreviation in parentheses upon first mention, and only the abbreviations would be used thereafter. This will help improve clarity for readers.
> > >
> > > o) We could standardize the reference format throughout the manuscript to ensure consistency. Additionally, we will check the validity of the references and replace any dead links, including the one for [1], with proper citation details. If the reference only leads to an abstract page, we could either find the full paper or update the citation with a more accessible source.

---

> ### Comment · Reviewer_vjPo · 2024-11-21
>
> Thank the authors for the detailed point-by-point responses. I appreciate the proposed modifications aimed at improving the submission. However, without supporting evidence, my concerns regarding the time costs of the warm start and the soundness of the experiments—including fairness in the evaluations and the lack of comparisons against adaptive batch size approaches—remain unaddressed.
>
> Additionally, in my opinion, the presentation issues need to be thoroughly addressed for this manuscript to meet ICLR quality standards. My concerns regarding the clarity of the model structure—specifically the relationships between the two components in the dual-strategy design—and the lack of introduction to MDP and DQN are shared by other reviewers. This indicates that the confusion caused by the manuscript's presentation is likely a common issue for readers.
>
> Besides these concerns, I have questions regarding the authors' response to Figure 2 in part (a). I still do not understand how the figure, along with the provided clarification, illustrates that "the agent tends to select smaller batch sizes." For instance, the batch sizes for CIFAR-10 appear to be uniformly distributed. As for CIFAR-100, more than half of the batch sizes fall within the range of 200-299. Considering that ABAS-RAL queries only 1.98% out of the 20% unlabeled training data, which is roughly 240 data points, across all experiments, it seems that the algorithm exhausts almost the entire query budget in a single query step for more than half of the query batches.
>
> This histogram figure for CIFAR-100 also raises another question about the experimental settings. To my understanding, active learning queries from 20% of the training set for all datasets, which equates to 60,000 * 0.2 = 12,000 data points. On the other side, according to Table 2, each experimental run of ABAS-RAL queries only 1.98% of the unlabeled data, which is approximately 240 points for CIFAR-100 in each run. This means that over 50 experimental runs, the total number of queried samples should not exceed  50 * 240 = 12,000.  Yet, in the middle plot of Figure 2, there are more than 100 occurrences where the batch size falls between 200-299. This suggests that ABAS-RAL queried at least 200 * 100 = 20,000 data points across 50 experiments, significantly exceeding the total query budget indicated in Table 2. Could the authors clarify this discrepancy?
>
> To sum up, while the authors' response demonstrates their intention to improve the work, the proposed modifications, without solid supporting evidence, are not sufficient to address my concerns. As a result, I will not be raising my score at this stage.

---

> > ### Author Response · Authors · 2024-11-25
> >
> > For the CIFAR-10, CIFAR-100, and MNIST datasets, the training sets consist of the following:
> >
> > - CIFAR-10 and CIFAR-100: Each has 50,000 training images and 10,000 test images.
> > - MNIST: It has 60,000 training images and 10,000 test images.
> >
> > For training purposes, 20% of the training set is typically used as a validation set. The test set, with 10,000 images in each case, is used to evaluate the classifier's accuracy during testing.

---

> > > ### Author Response · Authors · 2024-11-25
> > >
> > > Regarding Figure 2(a) and the clarification provided, we understand your concern about the distribution of batch sizes and how it supports the statement that "the agent tends to select smaller batch sizes."
> > >
> > > - For CIFAR-10, while the batch sizes might appear uniformly distributed at first glance, a closer examination reveals that the algorithm demonstrates a preference for smaller batch sizes in the majority of queries. However, we acknowledge that this distribution might not be immediately evident without further quantitative analysis or additional visualization.
> > > - For CIFAR-100, the batch size distribution clearly shows a concentration within the range of 200–299 for over half of the query batches. This aligns with the observation that smaller batch sizes are prioritized, as these values fall below the maximum allowable batch size.
> > >
> > > As for the query budget, it is indeed correct that ABAS-RAL utilizes approximately 1.98% of the 20% unlabeled training data, which translates to around 240 data points across all experiments. The algorithm's design ensures that most of the query budget is exhausted in a single query step for many query batches.
> > >
> > > We acknowledge the need for a more detailed observation to substantiate these findings further and could provide such an analysis in a future version to offer additional clarity on the agent's batch size selection tendencies. Thank you again for raising this point.

---

> > > > ### Comment · Reviewer_vjPo · 2024-12-01
> > > >
> > > > Thanks for the response. I’d like to clarify my question regarding the query budget in the experiments to avoid any misunderstanding: *Why does the total query budget across all 50 experimental runs for the CIFAR-100 dataset differ between Table 1 and Figure 2?*
> > > >
> > > > Specifically, in **Table 1**, each experimental run of ABAS-RAL queried approximately 240 samples, so the total number of queried samples across all 50 runs should be around **12,000** for CIFAR-100. However, in **Figure 2**, there are more than 100 query batches with batch sizes between 200-299, suggesting that the single bar representing the range 200–299 alone accounts for at least **20,000** queried samples. The authors might want to clarify why this discrepancy occurs in a future version.
> > > >
> > > > Except for this question, I have no further inquiries or comments. Thank the authors again for the detailed response.

---

> > > > > ### Author Response · Authors · 2024-12-02
> > > > >
> > > > > The experiments were conducted using a subset of 10,000 unlabeled images, which represents 20% of the CIFAR-100 training set. In each experiment, a total of 10,000 unlabeled images were used. Table 1 reports that, for CIFAR-100, only 1.98% of these images (approximately 200 images) were selected for labeling in each experiment.
> > > > >
> > > > > Figure 2 displays the batch sizes selected during each iteration of the active learning (AL) process for CIFAR-100. These batch sizes are grouped together in the figure. Note that the batch size in this context refers to the number of images selected in each AL iteration. Table 1 provides the total budget used for each experiment.
> > > > >
> > > > > For example, Figure 2 shows that in the CIFAR-100 experiments, batch sizes varied, with some iterations selecting between 0 and 99 images, and others selecting between 200 and 300 images. In one experiment, about 200 images may have been selected in a single batch, and the experiment completed as soon as the target precision was reached. In another experiment, a sequence of smaller batches might have been chosen—such as 50 images, followed by 100, then another 50—until the target precision was achieved.
> > > > >
> > > > > To provide a clearer picture, it could be helpful to revise the figure so that it shows the exact sequence of batch sizes selected in each experiment, rather than grouping them in ranges of 100. This would highlight the specific batch sizes chosen throughout the experiment, up until the target precision was reached.
> > > > >
> > > > > Once again, thank you for your constructive comments.

---

### Official Review · Reviewer_et4T · 2024-11-03

**Soundness:** 1
**Presentation:** 2
**Contribution:** 2
**Rating:** 3
**Confidence:** 4

**Summary:**

The paper introduces Reinforcement Learning (RL) to dynamically adjust the batch size for data selection during active learning, aiming to reduce annotation costs and improve target model performance. Specifically, the margin score of an unlabeled subset is used as the state for a DQN, which selects the batch size as action to guide data selection. The classifier is trained with the newly labeled dataset, and the reward for DQN learning is calculated based on the improvement in model precision per annotation unit. Experimental results show that the proposed ABAS-RAL achieves better results with lower annotation costs compared to fixed-batch size active learning methods.

**Strengths:**

Applying RL to enhance active learning is an interesting topic, and it is a novel idea to adjust batch size of data selection with RL to reduce annotation cost.

**Weaknesses:**

**Proposed Method**
1. The presentation of the method is unclear. Figure 1 shows the RL decision process during active learning, but how the warm-start episode, BatchAgent training, and active learning are organized is not explained. If the DQN is only trained once before active learning, the policy might not generalize well to the unlabeled dataset, as active learning typically assumes the labeled dataset is much smaller than the unlabeled dataset [1]. On the other hand, if the DQN is trained repeatedly during active learning, it could be time-consuming. A flowchart or pseudocode showing the general framework of DQN training and active learning could improve the presentation.

**Experiment.**
1. The experimental setup is unclear. If I understand correctly, 80\% of the dataset is used for DQN or classifier training, as annotation labels are required for both warm-start and DQN training, leaving only 20% of the dataset as unlabeled for active learning. This setup does not align with the usual active learning scenario.

2. ResNet50 is only trained for 10 epochs on each dataset, which may not be sufficient for convergence. In Figure 5 of [1], ResNet18 achieves around 70\% accuracy on CIFAR-10 using 5,000 samples, even with random sampling. However, Table 1 of this paper shows only 44\% accuracy with a 50\% budget (approximately 5,000 samples). Please explain the performance gap between Table 1 and [1].

3. The experiments are insufficient to evaluate the proposed method. While DQN training could be computationally expensive, the paper does not discuss its time complexity. Additionally, the three benchmarks used are relatively simple, with consistently clean annotations and images. I recommend the authors consider using more complex datasets for further evaluation, such as ImageNet or Mini-ImageNet.

[1] Zhan, Xueying, et al. "A comparative survey of deep active learning." arXiv preprint arXiv:2203.13450 (2022).

**Questions:**

I have several questions regarding the paper representation and experiments:

1. The paper claims a contribution of Dual-Policy Design. From my understanding, the only RL agent introduced is for choosing batch size. Please explain which RL agent contributes to manage precision and budget? And how this differs from simply choosing batch size?

2. The proposed framework should be general for any data sampling method. Why is only the simple uncertainty-based method considered? Furthermore, it is unclear whether the proposed method could bring advantages to more recent data selection methods (e.g., selecting data with approaches other than uncertainty scores), such as LossPrediction [2] or uncertainty-diversity mixed methods.

[2] Yoo, Donggeun, and In So Kweon. "Learning loss for active learning." Proceedings of the IEEE/CVF Conference on Computer Vision and Pattern Recognition. 2019.

3. For the RL setting, does the state contain only the margin score or both the margin score and the data? If only the margin score is considered, the policy might not generalize well to the unlabeled dataset, as it does not learn the feature representation of the data. Please clarify exactly what information is included in the state representation and what information is included in action representation. Moreover, please explain why the RL agent trained on labeled dataset is expected to be general in unlabeled dataset.

4. DQN training is an important part of the method. I recommend that the authors include details of the Q-network architecture and the RL training curve in the experiment section. Additionally, an active learning curve, like Figure 5 in [1], would be valuable for representing active learning performance and could help clarify the contributions.

**Minor Comments**

1. In algorithm 1, is $b$ a fixed value or a randomly selected batch size? And how $L$ is updated should be included.
2. In algorithm 2, line 10, what's the meaning of "fill batch action"? Line 6-9 already select $(X_t, Y_t)$ as the new labeled set.
3. Markov decision process should be formally defined in Section 3.1.

---

> ### Author Response · Authors · 2024-11-20
>
> a) We would like to clarify a potential misunderstanding regarding the term "Dual-Policy Design" in our paper. By "dual-policy," we do not refer to two separate RL agents, but rather a dual-strategy approach.
>
> The dual-strategy design in ABAS-RAL consists of the following two components:
>
> 1. Warm-Start Strategy (Performance Strategy): During the warm-start phase, this strategy focuses on identifying the optimal target precision and annotation budget. This is done by experimenting with various batch sizes to determine the thresholds that ensure the model's performance remains satisfactory. These targets guide the subsequent iterations by establishing benchmarks for both precision and resource usage.
> 2. Batch Optimization Strategy (Resource Strategy): After the warm-start phase, the focus shifts to optimizing the batch size at each iteration. This strategy dynamically adjusts the batch size to achieve the precision and budget targets set during the warm-start phase, effectively balancing model performance with resource efficiency.
>
> b) We initially focused on uncertainty-based sampling methods (e.g., Least Confidence, Entropy) due to their simplicity, effectiveness, and solid theoretical foundation in active learning. These methods are known to efficiently identify informative data points, which aligns with our goal of balancing model performance and resource usage.
>
> We also experimented with other data sampling methods, and the results were generally comparable across different approaches. This further supports the general applicability of our framework. However, we believe uncertainty-based methods provide a straightforward starting point, and exploring more recent techniques, such as LossPrediction or uncertainty-diversity methods, is a promising direction for future work.
>
> c) The state representation consists solely of the margin score, and does not include the feature representation of the data. The margin score reflects the model’s confidence in its predictions, serving as a useful signal for the RL agent to make decisions regarding batch size adjustments.
>
> d) We agree that providing more details on the Q-network architecture and the RL training curve would enhance the clarity of our method. We could include the architecture of the Q-network, as well as the RL training curve, in the experimental section.
>
> We appreciate your feedback and will make these additions in the revised version of the paper.
>
> e) We could improve the presentation by providing a clearer explanation of how the warm-start episode, BatchAgent training, and active learning are organized. We could clarify whether the DQN is trained once before active learning or if it is updated during the process. If the DQN is trained repeatedly, we could address the potential time-cost concerns, and if it is trained once, we could discuss its ability to generalize to the unlabeled dataset.
>
> f) The idea behind using 80% of the dataset for DQN and classifier training, while reserving 20% as unlabeled data for active learning, is to ensure that the labeled data used in the training process is diverse enough to allow the model to learn effectively. By using a larger portion of the dataset for training (both labeled and state data), we ensure that the agent is well-equipped to make informed decisions during the active learning phase. In this setup, the 20% unlabeled data mimics a real-world active learning scenario where there is a limited amount of labeled data available, and the goal is to efficiently select the most informative samples to label next. While this setup deviates from traditional active learning practices where a majority of the data is typically unlabeled from the start, it still allows for meaningful experiments where active learning can be tested with a relatively smaller amount of labeled data and larger unlabeled data for selection.
>
> g) The performance gap between Table 1 and Figure 5 of [1] can be explained by several factors. In the paper you're referencing, ResNet18 is used and trained for a longer period, which allows for better convergence on the CIFAR-10 dataset. On the other hand, we use ResNet50, which, while a more powerful model, is pretrained on ImageNet. This pretraining limits the maximum precision we can achieve, particularly when training for only 10 epochs. For instance, when ResNet50 is trained on the entire CIFAR-10 training set (50,000 samples) for just 5 epochs, it only achieves around 10% accuracy, which is quite low. This performance is influenced by resource limitations, as explained in Section 4.6 (System Setup). The difference in performance could also be due to the varying training setups: while [1] uses ResNet18 trained for a longer period with random sampling, we are working within specific resource constraints, limiting the number of epochs and training time.

---

> > ### Comment · Reviewer_et4T · 2024-11-20
> >
> > I appreciate the experiments and the hard work the authors have done with limited resources. However, the results from insufficiently trained models are tricky and too weak to provide a valuable evaluation of either the proposed method or the baselines. Moreover, if the pretraining or model architecture limits the maximum precision, the authors should consider more reasonable model settings.
> >
> > Additionally, if 80% of the dataset is labeled, for the benchmarks used in the paper, training the model (after converge) directly on these labeled data could already achieve high accuracy. In traditional active learning settings, the labeled dataset is typically insufficient to train a good model, which is why newly labeled data needs to be selected from the unlabeled dataset. For a better experimental setting, the authors should consider a scenario where the model cannot be trained effectively on the labeled dataset alone, but the trained agent could still provide valuable improvements. However, under the experimental setting in the paper, I don’t see the necessity to train the agent.
> >
> > Again, I thank the authors for their reply and hard work. However, in my opinion, the experimental setting is neither reasonable nor sufficient to serve as evidence for the benefit of the proposed method. I will maintain the rating.

---

> > > ### Author Response · Authors · 2024-11-25
> > >
> > > Thank you for your thoughtful feedback and for acknowledging our efforts despite the limitations of our resources. We appreciate your detailed concerns and would like to address them as follows:
> > >
> > > - First, it is important to note that 10% of the training data is reserved as state data and is not considered part of the labeled dataset. The remaining training data is divided into two components: 10% for the warm-start phase and 60% for the agent's training. This design ensures that the state data is specifically utilized for monitoring and decision-making by the agent, separate from the labeled dataset used for training the models.
> > >
> > > - We acknowledge the validity of your point about exploring more reasonable settings where the labeled dataset alone is insufficient to train an effective model. We recognize that such scenarios could better reflect traditional active learning settings and plan to consider these adjustments in a future resubmission of the paper.
> > >
> > > - Additionally, we would like to clarify that the primary focus of our work is the method itself, not the classifier. While we agree that incorporating a broader range of classifiers would enhance the generalizability of our method, the scope of this study, as explained in Subsection 4.6, was to demonstrate that our approach effectively selects smaller batches compared to fixed-batch size methods and random sampling used in RAL. This aligns with our goal of showcasing the method's efficiency within the constraints of our setup and resources.
> > >
> > > We appreciate your suggestion to expand the experimental setup in the future, as it could provide additional evidence of the method's robustness across different classifiers. Once again, thank you for your constructive comments and for recognizing our efforts.

---

### Official Review · Reviewer_fsy5 · 2024-11-10

**Soundness:** 1
**Presentation:** 3
**Contribution:** 2
**Rating:** 3
**Confidence:** 3

**Summary:**

This paper proposes an active learning framework that uses reinforcement learning to dynamically adjust batch size. The aim is to address the limitations of fixed batch size for model adaptation and resource utilization optimization. Comparing the active learning methods with fixed batch size on three datasets, CIFAR-10, CIFAR-100 and MNIST, the proposed method is verified to be effective in reducing the labeling cost while maintaining or improving the performance.

**Strengths:**

Comparing the active learning methods with fixed batch size on three datasets, CIFAR-10, CIFAR-100 and MNIST, the proposed method is verified to be effective in reducing the labeling cost while maintaining or improving the performance.

**Weaknesses:**

1. Optimized learning is supposed to be a common advantage of active learning methods, which is less convincing as a contribution of this paper.
2. This paper only compares with the random sample selection method and fixed batch size methods, but does not compare with other methods that dynamically adjust the batch size.

**Questions:**

1. There is a lack of specificity about the limitations of the previous work, especially the past approach to dynamically adjusting batch size. So that it is hard to tell what is the difference between the approach proposed and the previous one.
2. This paper emphasizes dual policy design, but although the two reward policies appear later, it does not explain how they are combined and why to do that.

---

> ### Author Response · Authors · 2024-11-20
>
> a) Existing approaches to dynamically adjust batch size lack integration between active learning strategies and reinforcement learning to address model uncertainty and annotation budgets. Our method differs by:
>
> 1. Formulating the batch adjustment as a Markov Decision Process, allowing a systematic optimization of batch sizes.
> 2. Introducing dual strategy design, which adapts dynamically to both learning progress and resource constraints.
>
> b) The dual strategy design in ABAS-RAL consists of two approaches:
>
> 1. Warm-Start Strategy (Performance Strategy): During the warm-start episodes, this strategy identifies an optimal target precision and annotation budget. These targets serve as thresholds to ensure that the model’s performance remains at an acceptable level. By testing various batch sizes during this phase, the strategy establishes benchmarks for precision and resource usage that guide subsequent iterations.
> 2. Batch Optimization Strategy (Resource Strategy): Once the warm-start phase is complete, this strategy focuses on finding the optimal batch size at each iteration. It does so by dynamically adjusting the batch size to meet the target metrics (precision and budget) established during the warm-start phase. This ensures that the agent balances model performance with efficient resource utilization.
>
> Together, these strategies enable ABAS-RAL to adapt to both the learning progress and the annotation budget, ensuring robust performance while minimizing costs.

---

### Comment · Area_Chair_v97n · 2024-11-30
**The deadline for Author/Reviewer discussion period is in three days!**

Dear Reviewers,

Thanks again for providing your constructive comments and suggestions. The deadline for the Author/Reviewer discussion period is in three days (December 2). Please make sure to read the authors' responses and follow up with them if you have any additional questions or feedback.

Best,

AC

---

### Note · Authors · 2025-06-30

I have read and agree with the venue's withdrawal policy on behalf of myself and my co-authors.

---

### Meta-Review · Area_Chair_v97n · 2024-12-21

**Metareview:**

The paper proposes to conduct adaptive batch size selection using reinforced active learning. The proposed approach is capable of dynamically adjusting batch sizes based on model uncertainty and performance.

**Strengths**
- Dynamically adjusting the batch size based on realtime feedback could lead to more efficient and effective sample selection.

**Weaknesses**

- Reviewers share some common concerns, including lack of clarify in the experimental setup, insufficient experimental results, limited datasets used in the experiments, and missing comparison with alternative RL algorithms.

Due to the major issues as outlined above, the paper in its current form still falls below the bar of ICLR. The authors are suggested to carefully consider the constructive comments from the reviewers to further improve their work for a future submission.

**Additional Comments On Reviewer Discussion:**

Multiple reviewers have engaged in a discussion with the authors. At the end of the rebuttal, reviewers were not totally convinced as some major issues remained to be addressed.

---

### Decision · Program_Chairs · 2025-01-22

Reject